# Effects of Fermentation with *Tetragenococcus halophilus* and *Zygosaccharomyces rouxii* on the Volatile Profiles of Soybean Protein Hydrolysates

**DOI:** 10.3390/foods12244513

**Published:** 2023-12-18

**Authors:** Chenchen Cao, Geoffrey I. N. Waterhouse, Weizheng Sun, Mouming Zhao, Dongxiao Sun-Waterhouse, Guowan Su

**Affiliations:** 1School of Food Science and Engineering, South China University of Technology, Guangzhou 510640, Chinag.waterhouse@auckland.ac.nz (G.I.N.W.); femmzhao@scut.edu.cn (M.Z.);; 2Guangdong Food Green Processing and Nutrition Regulation Technologies Research Center, Guangzhou 510650, China; 3School of Chemical Sciences, The University of Auckland, Private Bag 92019, Auckland 1010, New Zealand

**Keywords:** soybean protein hydrolysates, lactic acid bacteria, yeast, volatile flavor compounds, GC-MS, GC-IMS

## Abstract

The effects of fermentation with lactic acid bacteria (LAB) and yeast on the aroma of samples were analyzed in this work. The volatile features of different soybean hydrolysates were investigated using both GC-MS and GC-IMS. Only 47 volatile flavor compounds (VFCs) were detected when using GC-IMS, while a combination of GC-MS and GC-IMS resulted in the identification of 150 compounds. LAB-yeast fermentation could significantly increase the diversity and concentrations of VFCs (*p* < 0.05), including alcohols, acids, esters, and sulfurs, while reduce the contents of aldehydes and ketones. Hierarchical clustering and orthogonal partial least squares analyses confirmed the impact of fermentation on the VFCs of the hydrolysates. Seven compounds were identified as significant compounds distinguishing the aromas of different groups. The partial least squares regression analysis of the 25 key VFCs (ROAV > 1) and sensory results revealed that the treatment groups positively correlated with aromatic, caramel, sour, overall aroma, and most of the key VFCs. In summary, fermentation effectively reduced the fatty and bean-like flavors of soybean hydrolysates, enhancing the overall flavor quality, with sequential inoculation proving to be more effective than simultaneous inoculation. These findings provided a theoretical basis for improving and assessing the flavor of soybean protein hydrolysates.

## 1. Introduction

Soybean protein, including soybean protein concentrate and soybean protein isolate, is primarily derived from defatted soybean, offering an economical and efficient source of dietary protein [1]. Due to its balanced amino acid profile and significant nutritional value, soybean protein is widely regarded as an excellent substitute for animal proteins. Soybean protein are subjected to further processing (e.g., chemical/enzymatic hydrolysis, microbial fermentation, thermal treatments, or gastrointestinal digestion) to produce protein hydrolysates for various food applications [2]. Soy protein hydrolysates can exhibit beneficial physiological properties (e.g., solubility, emulsifying capacity, and digestibility), functional attributes (e.g., antihypertensive and antioxidative activities), and flavor characteristics (umami taste). However, some drawbacks of soybean protein hydrolysates, including the naturally occurring grassy or beany flavor of soybean and potentially increased bitter taste, greatly restrict their further food applications considering consumer acceptance [3]. Accordingly, there has been an ongoing effort from researchers to develop novel processes or to optimize processing conditions for improving the flavor profiles of soybean protein hydrolysates. 

Some literature reported that an appropriate enzymatic hydrolysis could improve the taste of soybean hydrolysates (e.g., increase in umami taste and reduction in bitterness) for food seasoning [4]. A range of commercial enzymes has been demonstrated to effectively reduce the bitterness of soybean protein hydrolysates [5]. In terms of odor, the prevailing methods to mitigate undesirable aromas in soybean protein hydrolysates include physical treatments, chemical processes, activated carbon adsorption, and fermentation [6]. In our previous studies, fermentation was demonstrated to effectively elevate the contents of umami amino acids, sweet amino acids, and organic acids in soybean hydrolysates, thereby enhancing the taste profiles, including umami and sweetness [7].

Moreover, fermentation has been widely used to improve the aroma of foods, including grains and legumes, through the biochemical reactions, enzymatic actions, and metabolic processes of the microorganisms [8,9]. Lactic acid bacteria (LAB) can break down protein, fat, and carbohydrates to produce flavor compounds, including aldehydes, alcohols, esters, and acids, as LAB are capable of producing enzymes (like protease, lipase, esterase, reductase, and peroxidase) during fermentation to hydrolyze proteins into peptides and free amino acids [10,11]. Besides producing flavor precursors and offering probiotic properties, LAB can also convert the food components in raw materials to biologically active substances, such as bioactive peptides and phenolic metabolites [12]. Some halophilic aromatic yeasts (such as *Zygosaccharomyces rouxii*, *Candida versatilis*, *Candida etchellsii*, and *Pichia guilliermondii*) are other types of microbial strains for the fermentation of soybean products and give soybean fermented products distinct flavors and a special texture [13]. During fermentation, yeasts can modify carbohydrates and convert them to various flavor-active metabolic by-products and also facilitate the biosynthesis of esters, alcohols, and other volatile compounds [14]. Nevertheless, relatively few studies have comprehensively analyzed how improvements in fermentation affect the aroma of soybean hydrolysates.

Gas chromatography-mass spectrometry (GC-MS) is advantageous based on its wide detection range, high separation efficiency, and high capacity for identifying unknown compounds based on the National Institute of Standards and Technology (NIST) database [15]. However, a complex preconditioning procedure is required prior to GC-MS analysis, and it also has a relatively high cost due to the required vacuum and helium as the carrier gas. Recently, gas chromatography-ion mobility spectrometry (GC-IMS) has also been used for analyzing the flavors of foods, such as grains, edible mushrooms, meats, fishes, alcohol, tea, and fruits [16]. This technique combines the advantages of GC (high separation capacity) and IMS (fast response and high sensitivity), offering a rapid, accurate, simple, and intuitive analysis [17]. GC-IMS equipped with an automatic headspace injector allows for the detection of volatile compounds from the samples without any preprocessing and multiple organic reagents. However, few experimental studies have been performed to investigate the volatile profiles of soybean hydrolysates using GC-IMS or the combination of GC-IMS and GC-MS. 

This study attempted to investigate the effects of fermentation by *Tetragenococcus halophilus* and *Zygosaccharomyces rouxii* on the volatile profiles of soybean protein hydrolysates. GC-MS and GC-IMS were employed as identification methods to achieve a better examination of aroma profiles in different soybean hydrolysates. Orthogonal partial least squares analyses (OPLS-DAs) and partial least squares regression (PLSR) were used to further assess the effects of the fermentation on the aroma profiles of the obtained soybean hydrolysates and explain the sensory properties with volatile compounds. The results of this research could guide the process design for producing soybean protein hydrolysates with an improved volatile flavor compound (VFC) profile and demonstrate the capacities of GC-IMS and its combination with GC-MS for analyzing the VFCs of soybean protein hydrolysates.

## 2. Materials and Methods 

### 2.1. Materials

Defatted soybean flour was provided by Shandong Yuwang Industrial Co., Ltd. (Yucheng, Shandong, China). The commercial proteases (Flavourzyme 500 MG and Alcalase 2.4 L) were purchased from Novozymes Biotech (Tianjin, China). 2-Methyl-3-heptanone, as an internal standard in this study, was purchased from Sigma- Aldrich (St. Louis, MO, USA). The *n*-alkanes (C7-C20), for linear retention index (RI) determination, were obtained from SUPELCO (Bellefonte, PA, USA). The solvents were of chromatographic grade.

### 2.2. Microorganisms and Growth Conditions

*Tetragenococcus halophilus* (CICC 10286) and *Zygosaccharomyces rouxii* (CICC 1378) were provided by the China Center of Industrial Culture Collection (Beijing, China). The cultivation conditions for both strains were slightly modified according to a previous study [9]. *T. halophilus* 10286 strains were cultured in MRS broth medium with 10% (*w*/*v*) NaCl at 30 °C for 48 h. Yeast strains were grown in liquid yeast extract peptone dextrose (YPD) medium at 30 °C (160 rpm, 48 h). 

### 2.3. Preparation of Fermented Hydrolysates of Defatted Soybean Flour 

Soybean protein enzymatic hydrolysate was prepared using the method of Zhang et al. (2016) [18] with slight modifications. The defatted soybean flour and deionized water were mixed (1:7, *w*/*v*) and subjected to sterilization (121 °C, 15 min). Flavourzyme (0.5%) and Alcalase (1%) were added after the sterilized defatted soybean water suspension had reached room temperature. The mixture was incubated at 55 °C for 18 h. Then, inactivation of the enzymes was conducted at 95 °C for15 min. The resulting hydrolysates were centrifuged at 8000× *g* at 4 °C for 25 min, and the supernatant was collected and stored at −20 °C. Then, the resulting hydrolysate was divided into three groups: (1) SH-0, the control without further inoculation of any microbe; (2) SH-1, co-inoculated with *T. halophilus* 10286 (1 × 10^8^ CFU/mL) and *Z. rouxii* 1378 (1 × 10^7^ CFU/mL) followed by incubation at 30 °C for 8 d; (3) SH-2, initially inoculated with *T. halophilus* 10286 (1 × 10^8^ CFU/mL) and incubated at 30 °C until the pH decreased to 5.5. Subsequently, *Z. rouxii* 1378 (1 × 10^7^ CFU/mL) was added at 30 °C. The total fermentation time was 8 days. The samples of all of these groups were centrifuged at 8000× *g* at 4 °C for 30 min. The resulting supernatant was filtered through a 0.22 μm membrane filter, and the filtrate was then stored at −20 °C. 

### 2.4. GC-MS Analysis

The volatile compounds in all the obtained soybeans hydrolysate samples were analyzed following a published procedure [13] with slight modifications. The GC-MS analysis was carried out using a Trace DSQ-II instrument (Thermo fisher, Waltham, MA, USA) equipped with a mass selective detector (ISQ, Thermo Science, Waltham, MA, USA). 

Briefly, each of the samples (5 g) and 20% NaCl were transferred to a 20 mL headspace vial, and then, an aliquot (20 μL) of 2-methyl-3-heptanone (concentration: 17.6 mg/L) was added as an internal standard. Headspace solid-phase microextraction (HS-SPME) extraction was performed using the preconditioned extraction fiber needle (50/30 μm, CAR/PDMS/DVB). The headspace vial was equilibrated at 45 °C for 20 min; then, the aged head of the extraction fiber needle was inserted into the head space of the sample vial for 30 min. A clean-up step was accomplished by heating the extraction head at 270 °C for 10 min, to prevent cross-contamination between consecutive analyses. 

The volatile compounds were analyzed on a TR-5MS capillary column (60 m × 0.32 mm × 0.5 μm, Thermo Scientific, Waltham, MA, USA). The carrier gas was helium at a constant flow rate of 1 mL/min. The oven temperature was initially maintained at 40 °C for 6 min, then raised to 120 °C at a rate of 4 °C/min, and finally increased to 280 °C at 10 °C/min and kept at 280 °C for 15 min. The temperatures of both the ion source and transfer line were set at 250 °C. The mass detector was operated in the electron impact mode. An ionization energy of 70 eV and a 35–350 mass unit range were used. The volatile compounds were identified by comparing their linear retention index (RI) values and mass spectra with those of the alkane standard solutions (C7–C20) and the mass spectra in the NIST database (v14.0) or Wiley library. The concentrations of volatile compounds were determined using the method reported by Zhao et al. [19]. All analyses were performed in triplicate.

### 2.5. GC-IMS Analysis 

The volatile compounds in the test samples were further analyzed using an IMS instrument (FlavourSpec^®^, Dortmund, Germany) equipped with an auto sampling unit and Agilent 490 gas chromatograph (Agilent Technologies, Palo Alto, CA, USA) with an MXT-5 capillary column (15 m × 0.53 mm ID, 1 μm FT), according to a published method [20] with some modifications. Briefly, 1.0 g of the soybean protein hydrolysate sample was placed into a 20 mL headspace vial for an incubation at 50 °C for 15 min. Subsequently, 300 μL of headspace gas was automatically injected into the injector (500 rpm; splitless mode) using a heated syringe at 85 °C. 

The volatile compounds were separated with the MXT-5 capillary column (under 50 °C isothermal conditions) using nitrogen (99.999% purity) as the carrier gas at a programmed flow: 2 mL/min for 2 min, raised to 100 mL/min within 10 min, increased to 150 mL/min within 10 min, until the flow stopped. Volatile compounds were identified by comparing the RI and the drift time with those in the GC-IMS library and NIST database. The results were expressed as “an average relative content of each volatile compound to that of total aroma content” [17]. The obtained IMS data were subjected to chemometric analysis using the instrumental software VOCal 0.4.03, which includes plugins Laboratory Analytical Viewer (LAV), GC-IMS Library Search, and the Reporter and Gallery plot plugins. Each group was analyzed in triplicate.

### 2.6. Sensory Evaluation

Ten experienced panelists (six females and four males, aged 20–28 years) from South China University of Technology (Guangzhou, China) were invited to evaluate the odors of soybean hydrolysate samples in the laboratory (23 ± 2 °C), which complied with international standards (ISO 8589: 2007), under normal lighting. These sensory assessors received sensory training according to ISO 8586:2023, the international standard approach for training and supervising assessors, and possessed over one year of experience in evaluating the sensory characteristics of various food samples. The procedures for participant recruitment and consent acquisition were rigorously conducted in accordance with the ethical guidelines in the Declaration of Helsinki. Informed consent was obtained from each assessor before starting the sensory evaluation. Odor attributes evaluated in this study included the sour, fatty, aromatic, beany, caramel, and overall aroma. A seven-point scale was utilized to measure the intensity of each characteristic, where 1 signified a low intensity and 7 represented a high intensity [21].

### 2.7. Statistical Analysis

Data were evaluated using SAS software (version 9.3, SAS Institute Inc., Carrey, NC, USA). Duncan’s multiple range tests and one-way analysis of variance (ANOVA) were performed to determine any significant differences (*p* < 0.05) among the different samples. The hierarchical clustering analysis was performed using Origin pro software (version 2021b, USA). OPLS-DA and PLSR analyses were conducted to further evaluate the characteristics of samples using SIMCA 14.1.

## 3. Results and Discussion

### 3.1. Volatile Compounds Identified in Soybean Protein Hydrolysate Samples via GC-MS 

Typically, odor is one of the most important factors in accepting a product from the consumers perspective. The relationship between volatile compounds and odor descriptions has been well established. Thus, the identification of volatile compounds can predict the odor profile of each soybean protein hydrolysate obtained and determine the differences in volatile profiles among these hydrolysate samples. In the present study, a total of 127 VFCs were identified via SPME-GC-MS, with 86in SH-0, 108 in SH-1, and 104 in SH-2, respectively (Figure 1A). The Venn diagrams indicated that the presence of fermentation with the LAB (*T. halophilus* 10286) and yeast (*Z. rouxii* 1378) greatly increased the number of species of volatile flavor compounds (compared with SH-0). Also, the number of species of volatile flavor compounds was affected by the method of adding the LAB and yeast for fermentation, simultaneous inoculation of the LAB, and yeast or sequential addition–incubation of the LAB and yeast. It has been reported that the flavor of fermented foods may be changed due to the generation of new substances or a change in the concentrations of VFCs during fermentation [22]. The distinct flavor of fermented bean products is determined mainly by some species of volatile compounds, such as alcohols, acids, esters, ketones, and aldehydes. In this study, 11 types of volatile compounds were generated during fermentation, including alcohols (23), aldehydes (17), acids (10), esters (11), ketones (21), phenols (3), sulfurs (6), pyrazines (19), furan and pyran (8), pyridine and pyrrole (3), and others (6). Figure 1B shows the changes in the concentrations of the 11 types of compounds in the different soybean hydrolysates. Cluster analysis results indicated that alcohols, acids, esters, phenols, sulfur-containing compounds, pyridine, and pyrrole, as well as furan and pyran, clustered into group I. Meanwhile, aldehydes, ketones, pyrazines, and others were categorized into group II. Compared with those in SH-0, the concentrations of the identified compounds of group I in the samples fermented with the LAB and yeast increased significantly, whereas the concentration of group II compounds exhibited a notable decrease compared to that in the control samples (*p* < 0.05). Based on the relative contents of VFCs (Figure 1C), it was found that aldehydes, ketones, and pyrazines were predominant in the control groups. In SH-0, ketones were the most abundant species (accounting for 42.6%), followed by aldehydes (38.40%). Nevertheless, in SH-1/-2, alcohols had the highest proportion, followed by aldehydes. These results confirmed that the fermentation with the LAB and yeast greatly altered the proportions of the VFCs in the soybean hydrolysate samples. 

Hierarchical clustering was used to help detect the patterns of the GC-MS data to elucidate the relationships between the various VFCs and the preparation methods of soybean hydrolysate samples [12]. The red and blue areas of Figure 2 represent the volatile flavor compounds at higher and lower concentrations, respectively. Two obvious clusters are shown in Figure 2. The first cluster was SH-0, which had higher proportions of aldehydes (e.g., 3-methyl-butanal, 2-methyl-butenal, and heptanal), and ketones (e.g., 2-butanone, 4-methyl-2-pentanone, and 3-methyl-2-pentanone). The other cluster included SH-1 and SH-2, which had higher proportions of alcohols (benzyl alcohol, 2-heptanol, and 1-pentanol), acids (acetic acid, 3-methyl-butanoic acid, and 2-methyl-propanoic acid), and esters (phenethyl acetate, ethyl acetate and isoamyl acetate). These results were consistent with the finding of previous research [23] that fermentation with LAB and yeast could significantly improve the formation of VFCs.

All the identified compounds are listed in Appendix A. In general, the analyzed soybean hydrolysate samples differed in the number of species and concentrations of alcohol compounds. The alcohol compounds are mainly produced via the oxidation and degradation of the secondary hydroperoxides derived from polyunsaturated fatty acids, the degradation of amino acids, and the reduction from the corresponding aldehydes mediated by alcohol dehydrogenases [24]. While some straight-chain alcohols have minimal aroma, many alcohols (especially aromatic and aliphatic alcohols with relatively long chains) give aromas, such as menthol (peppermint), ethyl maltol (cooked fruit), and furaneol (strawberry) [13]. Among the detected 23 alcohol compounds, 3-methyl-1-butanol, 2-methyl-1-butanol, ethanol, and 1-octen-3-ol were the dominant compounds. Compared with SH-0, large increases in the concentrations of these alcohol compounds were found in the samples subjected to the fermentation with the LAB and yeast. Fermentation also caused small increases in the concentrations of 1-hexanol, prenol, and phenylethyl alcohol, which gave alcoholic, malty, floral, mushroom-like, and fruity aromas [13]. Alcohols with low odor thresholds are known as one type of key contributors to the flavor of fermented soybean hydrolysates, giving pleasant fruity and floral aromas. Interestingly, some new VFCs, including 2-methyl-1-propanol, 1-heptanol, 2-heptanol, and 4-methyl-1-pentanol, were produced after the fermentation. These compounds have been associated with green, nutty, and alcoholic flavors. Jiang et al. [14] reported that the concentrations of ethanol and phenylethyl alcohol were significantly higher in the fermented soy samples with yeasts added during fermentation, compared with those in the control group. Besides their direct contributions to product flavor (e.g., giving specific aromas), alcohols can make indirect contributions to product flavor by interacting with acids to produce esters, another important class of aroma compounds.

Aldehydes are mainly derived from lipid oxidation, degradation of precursor amino acids, and conversion of alcohols, giving pleasant aromas, such as grass, cheese, malt, and fruit flavors [25]. In this study, a total of 17 aldehydes were identified, with 2-methyl-propanal, 3-methyl-butanal, and 2-methyl-butanal being the main components. The concentrations of these aldehydes were abundant in the control samples, and fermentation with the LAB and yeast drastically decreased the concentrations of these aldehydes. These findings were in line with previous results [26]. Aldehydes are known as inherently unstable compounds and can easily be reduced to alcohols or oxidized to acids in food systems, particularly in the presence of microorganisms [27]. This might partially account for the decreased contents of aldehydes and the increased contents of alcohols for the fermented soybean hydrolysates. However, the concentrations of some aldehydes increased after the fermentation with the LAB and yeast, such as benzaldehyde (marzipan, almond, and nutty flavor), 3-methyl-2-butenal (sweet fruity, pungent, nutty with an almond and cherry nuance), hexanal (green, grassy), benzeneacetaldehyde (floral, fruity), and nonanal (beany, fatty, grassy). These aldehydes may be associated with the slight degradation of relevant amino acids and have low odor threshold values, thereby contributing greatly to the flavor of samples. Yi et al. [21] pointed out that the decreases in the concentrations of aldehydes in fermented mung bean might be beneficial for the overall aroma of the fermented foods, as high concentrations of aldehydes could cause undesirable flavors.

Acids are known to account for the sour flavor of foods. In this study, all the fermented soybean hydrolysates had a significant acidic smell, likely due to a large amount of acetic acid detected in these samples. Both the number of species and the concentrations of acid compounds increased greatly after the fermentation with the LAB and yeast, compared with those in the control group. Similar results were found in the fermentation of beans by *Lactobacillus plantarum* [21]. Among the identified 10 acid compounds, 2-methyl-propanoic acid (cheese), butanoic acid (cheese, fruity), and 3-methyl-butanoic acid (fruity) had significantly increased contents after the fermentation with the LAB and yeast, thereby contributing greatly to the sour flavor of the samples. Compared with the control samples, the concentrations of propanoic acid and octanoic acid also increased significantly, which might be associated with the high concentrations of alcohols available for further oxidation to acids [14]. Carbohydrate metabolism or lipid degradation in fermentation systems could lead to the production of acids.

Esters are usually generated due to the non-enzymatic esterification of alcohols and organic acids or enzymatic catalysis mediated by microorganisms. The esters produced by fermentation possess low flavor thresholds, thereby contributing greatly to fermented product flavor (mostly fruity and floral flavors) [28]. As shown in Appendix A, esters were the most prominent VFCs, in terms of the increases in the number of species and the concentrations of compounds. The total concentrations of esters in SH-1 (54.29 μg/L) and SH-2 (75.85 μg/L) were 22 and 31 times higher than that of SH-0 (2.45 μg/L), respectively. The formation of esters was mainly through the enzymatic esterification of fatty acids and alcohols mediated by microorganisms during fermentation [28]. In this study, 11 esters were detected, with ethyl acetate accounting for a high proportion, which was associated with the relatively high concentration of ethanol in the fermented samples. These results were in line with previous research [14] on the role of aroma-producing yeasts in the fermentation of soy sauce. Isoamyl acetate with a low odor threshold value was generated after the fermentation, giving a banana-like fruity flavor to the samples. Interestingly, isobutyl acetate, ethyl lactate, phenethyl acetate, and γ-nonanolactone were also detected in the soybean hydrolysates after the fermentation. Similar results were reported by Qi et al. [29]. The authors found that *W. versatilis* exhibited a high metabolic capacity and could facilitate the formation of esters via alcohol acetyltransferase and ester hydrolase.

Ketones are generated mainly through amino acid degradation or the beta-oxidation of unsaturated fatty acids and provide tallow and burnt aromas or a pleasant fruity, floral, and fragrance aroma (for ketones with relatively long carbon chains) [30]. Some ketones can be formed via the Maillard reaction and microbial metabolism [31]. In this study, a total of 21 ketones were detected in the soybean hydrolysate samples. The primary ketones were 2-butanone, acetoin, 2-heptanone, and isophorone, which gave fruity, cool wood, and cheese flavors. Furthermore, the concentrations of some ketones, such as methyl cyclopentenolone (sweet), acetophenone (flower and almond), and 2,6,6-trimethyl-2-cyclohexene-1,4-dione (tea and nutty), increased significantly after the fermentation with the LAB and yeast. Interestingly, β-damascenone was also detected in the fermented soybean hydrolysates. This compound has a low threshold and thus was anticipated to have an impact on the final flavor of the fermented hydrolysates. These results were consistent with the finding reported by Jiang et al. [14] that the concentrations of acetoin and acetophenone in the soy sauce fermented with yeasts were higher than those in the control group.

Phenols are generally produced from the degradation of lignins or the transformation of phenolic acids due to the actions of microorganisms or enzymes during fermentation [32]. This work indicated that the concentrations of phenols in SH-2 were higher than those in SH-0 and SH-1. In particular, the contents of 2-methoxy-phenol in the samples inoculated with the LAB and yeast were higher than those of the corresponding controls. It has been reported that 2-methoxy-phenol and 4-ethyl-2-methoxyphenol are characterized as having smoky aromas [33]. Furans are formed via the Maillard reaction, Strecker degradation, and thermal degradation of thiamine [34]. In this work, 2-furanmethanol was the main furan, while the contents of other furans were much lower. Notably, furfural and maltol were also detected soybean hydrolysates. Moreover, 2,5-dimethyl pyrazine (caramel), 2,6-dimethylpyrazine (coffee), and 2,3,5-trimethylpyrazine (potato) were the predominant pyrazines, which was consistent with a previous description that these pyrazines could typically be detected in fermented soybean products [35]. The relatively high concentrations of pyrazines were found in all the groups, suggesting that pyrazines play important roles in the flavors of the soybean hydrolysates. Pyrazines detected in the control groups were probably generated via the Maillard reaction. In addition, the concentration of methional significantly increased (*p* < 0.05) after fermentation. As reported, methional with a potato-like aroma is considered as an important volatile compound in Chinese soy sauce [22]. 

Fermentation is an important processing technique that reduces undesired flavors during the processing of cereals and legumes and produces unique fermented flavors. These impacts on product flavor are mainly attributable to the utilization of food components in raw material (as the substrates) and the biotransformation of certain metabolites mediated by microorganisms during fermentation, leading to the release of various flavor compounds or flavor-active substances [36]. In general, SH-2 exhibited higher contents of VFCs than SH-1, indicating a significant influence on the flavor of soybean protein hydrolysates when inoculated in sequence. It has been reported that the co-inoculation of *T. halophilus* and *Z. rouxii* during soy sauce fermentation can result in antagonism, with both experiencing varying degrees of growth inhibition [37].

### 3.2. Volatile Compounds in the Soybean Protein Hydrolysate Samples as Identified via GC-IMS

The differences in VFCs among the soybean hydrolysate samples were examined further via GC-IMS, which combined the high separation efficiency of gas chromatography with the fast response and high sensitivity of ion mobility spectrometry. The drift time is a useful parameter reflecting the mass and geometric structure of the ions of a chemical and can be determined based on the normalization of the ion migration time and reactive ion peak [20]. The data are presented through 3D and 2D topographical visualization in Figure 3. As shown in Figure 3A, the 3D-topographic plots of all soybean hydrolysates showed some similarities but also some differences. The majority of the signals were detected based on a retention time from 100 s to 800 s and drift time range of 1.0–2.0. The 2D-topographic plots (Figure 3B,C) also showed the significant differences in the corresponding signal intensity among the VFCs in the three groups of soybean hydrolysates. For the investigation of the aromatic component characteristics in several groups, the difference comparison model of the internal software was used. The topographic plot of SH-0 was selected as a reference and subtracted from the plots of other samples. With the retention time changing from 100 s to 800 s; scattered red areas were found in the two groups, suggesting that these VFCs had higher concentrations than those of the reference. The concentrations of quite a few VFCs in SH-1 and SH-2 were higher, compared with those in SH-0. These results suggested that the different soybean hydrolysates had different aroma profiles, and the fermentation with the LAB and yeast generally increased the concentrations of the volatile component in the hydrolysate samples. 

To further evaluate the differences in VFCs among the soybean hydrolysate samples, Gallery plot was used to draw fingerprint plots (Figure 3D). Each column represents a signal peak of a volatile compound in different hydrolysate samples, whilst each row represents all the signal peaks selected in a hydrolysate sample. Approximately four areas could be categorized to reflect the characteristics of the differential VFCs among different hydrolysate groups based on the color degree of each point and statistical analysis results of the VFC peak intensities. The fingerprints showed excellent reproducibility. A total of 47 VFCs were detected in all groups, among which 37 were successfully identified based on the GC-IMS library and NIST database. Several compounds displayed two peaks (which corresponded to their monomer and dimer forms), with the dimer eluted out after the monomer. The monomer and dimer were considered one compound in this investigation. These compounds could be separated into nine types: 8 alcohols, 11 aldehydes, 2 acids, 3 esters, 10 ketones, 1 sulfide, 1 pyrazine, 1 furan, and 10 other compounds. As shown by the red boxes in Figure 3D, higher concentrations of many VFCs were observed in the hydrolysate samples compared to those in SH-0, such as 2-methyl-1-propanol, pentan-2-ol, 3-methylbutan-1-ol, oct-1-en-3-ol, 3-methylthiopropanal, ethyl acetate, isoamyl acetate, 2-pentanone, and 3-hydroxybutan-2-one. All these VFCs gave specific aromas to the hydrolysate samples. However, as exhibited in the green boxes of Figure 3D, the relative contents of some VFCs decreased after the LAB-yeast fermentation, including 2-methylpropanal, 3-methylbutanal, hexanal, nonanal, 2-hexanone, and methyl isobutyl ketone. All the VFCs with decreased contents after fermentation were mainly aldehydes and ketones, which was consistent with the finding of previous research [21] that the contents of some aldehydes (nonanal, octanal, 2-furfural, and 3-methylbutanal) and ketones (acetophenone) in fermented mung beans decreased significantly or even disappeared. Such decreases in relative contents might be associated with the reduction reaction or decomposition induced by microbial fermentation. These results indicated that the LAB-yeast fermentation might be conductive to increasing the concentrations of VFCs in soybean protein hydrolysates. 

The identified compounds based on the RI and drift time are shown in Appendix A. Among the detected substances, aldehydes and ketones were the main compounds in control groups, whilst the primary components in the groups involving the LAB-yeast fermentation were alcohols, aldehydes, esters, and ketones. As depicted in Appendix A, the relative contents of the majority of alcohols increased after the fermentation with the LAB and yeast, such as 2-methyl-1-propanol, pentan-2-ol, 3-methylbutan-1-ol, and pentan-1-ol, thereby making high contributions to the aromas of samples. It was reported that the alcohols in soy sauce are generated primarily from sugars and amino acids during the fermentation under aerobic conditions, while some alcohols are produced through the reduction of the corresponding aldehydes mediated by yeasts [38]. In this study, the raw materials used for preparing the soybean protein hydrolysate samples were the same as those for making soy sauce. Moreover, soy protein hydrolysates could promote the growth of brewer’s yeast and drive fermentation, inducing faster reducing sugar consumption and higher ethanol generation [1]. This phenomenon might be associated with the relatively lower molecular weight and the more hydrophilic and electropositive amino acid residues brought in by the protein hydrolysate. Notably, the relative contents of oct-1-en-3-ol in SH-1 and SH-2 were higher than that of SH-0, which was in line with the result of GC-MS. Furthermore, the relative contents of 1-butanol in the treatment groups were lower compared with those in their corresponding control groups. These results might be related to the growth and metabolism of LAB and yeast and the possible transformation of acidic substances into esters [39]. This was further confirmed by the increase in butyl ester substances, such as butyl acetate and isoamyl acetate.

The relative contents of some aldehydes increased after the LAB-yeast fermentation, such as benzaldehyde and 3-methylthiopropanal. The generation of the majority of aldehydes possibly resulted from to the oxidation of unsaturated fatty acids. Previous studies reported that benzaldehyde and 3-methylbutanal could be produced due to the degradation of phenylalanine and leucine during the Maillard reaction or Strecker reaction [40]. However, the contents of 2-methylpropanal, 3-methylbutanal, (E)-2-methyl-2-butenal, (E, E)-2,4-hexadienal, nonanal, and hexanal decreased after fermentation. Similar results were investigated by Kewuyemi et al. [26], and the authors explained that the phenomenon was mainly related to the hydrolysis of proteins and the metabolite, which were the precursors of many volatile flavor substances. After fermentation, the aldehydes were further oxidized to acids or reduced to alcohols, which diminished the effect of the aldehydes.

Acid compounds were the major cause of acidic odors. Only two acids were detected via GC-IMS, which resulted in fewer species than those identified via GC-MS. Such a difference in the number of detected species between GC-IMS and GC-MS might be attributed to their detection principles. Most organic acids were less volatile, with only a few being highly volatile. GC-IMS is sensitive to those highly volatile acids. Due to their low relative contents and high odor thresholds, propanoic acid and 2-methylbutanoic acid imparted a minimal impact on the overall flavor of soybean hydrolysate samples. In terms of esters, only three species were detected, with the concentrations of ethyl acetate and isoamyl acetate increasing significantly after fermentation, thereby giving pleasant fruity and sweet aromatic odors. The fewer esters detected might be due to the low lipid content in soybean hydrolysates [35]. In this experiment, the relative contents of some ketones (including methyl isobutyl ketone and 2-hexanone) significantly decreased, while others increased, such as 2-pentanone (sweet fruity), 3-methyl-2-pentanone (cheese), and 2-heptanone (cheese and fruity aroma), compared with levels in the control groups (*p* < 0.05). These changes were consistent with the results of GC-MS. Furthermore, 3-methylthiopropanal (potato, tomato, and creamy), 2,6-dimethylpyrazine (nutty, roasted, meaty, coffee), and furfural (sweet, woody, baked bread) were also detected, which might be important contributors to the overall flavor of the soybean protein hydrolysates.

In summary, the major volatile flavors of soybean hydrolysates were derived from aldehydes and ketones. However, the contribution of aldehydes to the flavor of the final hydrolysate product was reduced significantly after fermentation, while the contribution of esters and alcohols increased. The aldehydes were oxidized to acids or reduced to alcohols, which decreased the effect of the aldehydes on samples. Similar findings were reported by Luelf, Vogel, and Ehrmann [35]; the type and concentrations of the volatile compounds, including alcohols, esters, acids, and furan, in the lupine fermented by *T. halophilus* and *Z. rouxii* increased significantly.

### 3.3. Analysis of VFCs Combining GC-MS with GC-IMS

Only 47 compounds were detected in the hydrolysate samples using GC-IMS alone, while 150 compounds could be detected by combining the two methods. The common VFCs in different soybean hydrolysate samples under different identification methods are shown in Venn diagrams (Appendix A). There were 22 to 26 volatile compounds detected simultaneously via GC-MS and GC-IMS. Most of the changes in VFCs detected using GC-IMS were consistent with the changing trends of the GC-MS data, including the increases in the contents of quite a few VFCs after the fermentation with the LAB and yeast. Especially, some flavor compounds detected only using GC-IMS were 1-butanol, pentan-2-ol, pentan-1-ol, (E)-2-methyl-2-butenal, (E, E)-2,4-hexadienal, hydroxyacetone, and 4-heptanone, which also made a key contribution to the overall flavors of samples. GC-IMS and GC-MS are known to operate based on different principles. The GC-IMS technique eliminates the need for enrichment, concentration, and other pretreatments before direct headspace injection, thus avoiding the difficult fragmented ion dissociation spectra [16,41]. IMS technology is particularly sensitive to compounds with a high proton affinity or electronegativity (like many volatile aroma compounds). In comparison, GC-MS requires more complicated preprocessing steps and more efforts to resolve fragmented peaks, as MS distinguishes ions according to the mass-to-charge ratio of the ions [42,43]. Quite often, GC-MS can identify the volatile compounds with relatively high contents, whilst GC-IMS can detect the volatile substances at low concentrations as this technique has a low detection limit. Therefore, the combined use of these two techniques allows for complementary analyses of VFCs, providing in-depth and comprehensive investigations on the characteristics and roles of various volatile compounds.

To further analyze the impact of different fermentation methods on the flavor of soybean protein hydrolysates, three OPLS-DA models were developed based on sensory evaluation, GC-MS, and GC-IMS results. The OPLS-DA score plots (Figure 4A,C,E) illustrated a distinct separation between the SH-0 and SH-1 groups, between the SH-0 and SH-2 groups, and between the SH-1 and SH-2 groups. These results indicated that there were significant differences in the contents of VFCs associated with different fermentation methods. In the three models, the R^2^X, R^2^Y, and Q^2^ were close to 1. Furthermore, a permutation test was conducted 200 times to validate that the model was not overfitted. This indicates that the model accurately represented the data distribution and demonstrated good explanatory and predictive capabilities. Moreover, the importance and explanatory power of each volatile compound was assessed by calculating the variable importance in projection (VIP) values from different models, aiming to screen key compounds that exert the most significant effect on aroma differences. A total of 20 variables with VIP > 1 and *p* < 0.05 were determined as differential VFCs (as shown in Figure 4B,D,F). Seven key differential compounds were consistently present in all three models, namely 3-methyl-1-butanol, ethanol, acetic acid, 2-methyl-1-propanol, 3-methyl-butanal, acetone, and 2-butanone. The differences in the contents of these VFCs might be the primary contributors to the aroma variations observed among the different soybean protein hydrolysate samples.

### 3.4. Analysis of Key Volatile Flavor Compounds

Aroma is a multifaceted expression resulting from the interplay of different flavor elements. It is vital to pinpoint which VFCs contribute to the recognized flavor characteristics. The final contribution of a VFC to the overall aroma is not solely dependent on its concentration but also its odor threshold. In this study, the relative odor activity value (ROAV) method was employed to investigate the aroma contributions of different compounds, with the specific calculation method referring to Yi et al. [18]. An ROAV greater than 1 indicates that the compound is a primary contributor to the sample flavor, and a higher ROAV signifies an increasing contribution of the compound to the overall flavor profile. Table 1 lists 25 VFCs (ROAV > 1) identified as key contributors to distinct aroma profiles. The results demonstrated that in the control group, the compounds with higher contributions were primarily aldehydes, including 2-methyl-propanal, 3-methyl-butanal, 2-methyl-butanal, hexanal, and nonanal. In contrast, the fermented groups exhibited a more diverse set of compounds with significant contributions, notably 1-octen-3-ol, 2-methyl-1-butanol, benzaldehyde, ethyl acetate, β-damascenone, 3-methylthiopropanal, and 2-pentanone. These substances give soybean protein hydrolysates fruit, caramel, malt, wine, mushroom, cheese, almond, honey, grass, fat, and meat flavors, which are important compounds of overall aroma.

To further assess whether the key flavor compounds reflected the flavor characteristics of the fermented soybean protein hydrolysates, 25 key VFCs and sensory results were analyzed using partial least squares regression (PLSR) analysis, and the data were preprocessed using unit variance (UV) scaling. As depicted in Figure 5, the model was successfully established with R^2^X = 0.992, R^2^Y = 0.952, and Q^2^ = 0.846. The model’s validity was further confirmed based on 200 permutation tests. Consequently, this model effectively elucidates the correlation between the key VFCs and sensory results. Figure 5B indicates that the control group was predominantly positively correlated with beany and fatty flavor, with strong correlations attributed to compounds, such as hexanal, nonanal, and 3-methyl-butanal. In contrast, the treatment groups displayed positive correlations with aromatic, caramel-like, sour, overall aroma, and the majority of key VFCs. Notably, SH-2 exhibited a stronger correlation than SH-1. Figure 5C shows that a total of 16 compounds (VIP > 1, *p* < 0.05) were identified as key differential compounds. Based on the key VFCs, Figure 5D predicted the overall aroma of different groups, showing that both fermentation methods significantly enhanced the overall aroma. In summary, fermentation decreased the fatty and bean-flavors of the soy protein hydrolysates, elevating its overall flavor profile, with sequential inoculation proving to be more effective than simultaneous inoculation.

## 4. Conclusions 

In conclusion, this study showed the feasibility of improving the aroma profiles of soybean protein hydrolysates through fermentation with *T. halophilus* and *Z. rouxii*. The combination of GC-MS and GC-IMS provided a comprehensive and effective analysis of VFCs, revealing substantial changes in their composition and concentration post-fermentation. LAB-yeast fermentation could increase the contents of alcohols, acids, esters, and sulfurs while decrease the contents of aldehydes and ketones. A total of 25 VFCs (ROAV > 1) were identified as key contributors for explaining the distinct aroma profiles among the three soybean protein hydrolysates. Notably, LAB-yeast fermentation not only diversified the array of key volatile flavor compounds but also shifted the flavor profile from predominantly beany and fatty to more desirable attributes, like fruity, caramel, malt, and wine notes. Particularly, sequential inoculation emerged as a more effective approach in enhancing the overall aroma, as evidenced by the PLSR analysis. These findings offer novel insights into aroma enhancement strategies for soybean protein hydrolysates, highlighting the potential of fermentation for improving the sensory qualities of food products. Future research may investigate the relationship between the volatile compounds and microbial metabolism.

## Figures and Tables

**Figure 1 foods-12-04513-f001:**
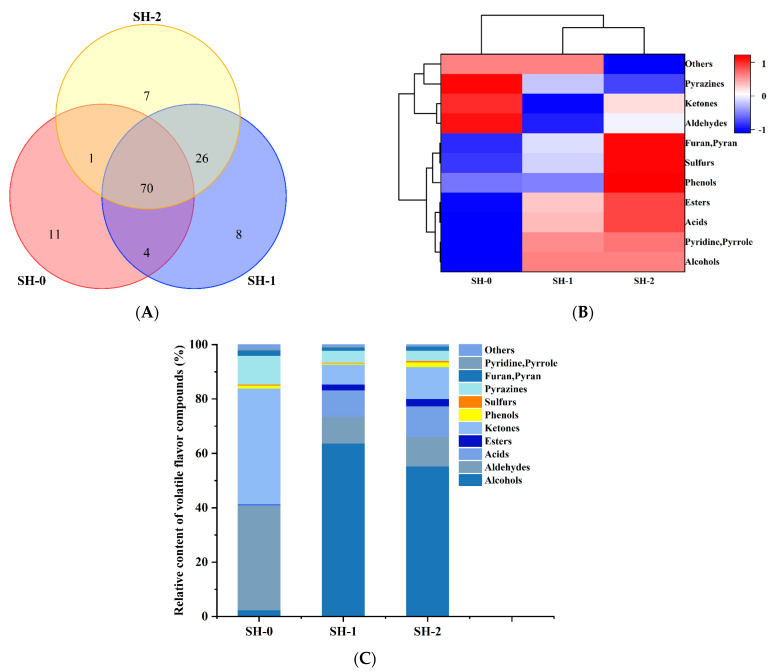
Volatile flavor compounds in the soybean protein hydrolysates obtained via different processes (**A**) Venn diagram; (**B**) cluster heatmap of different categories of volatile flavor compounds based on the GC-MS data after standardization. Darker red and darker blue, respectively, indicate higher and lower concentrations of volatile flavor compounds; (**C**) changes in the relative contents of volatile flavor compounds.

**Figure 2 foods-12-04513-f002:**
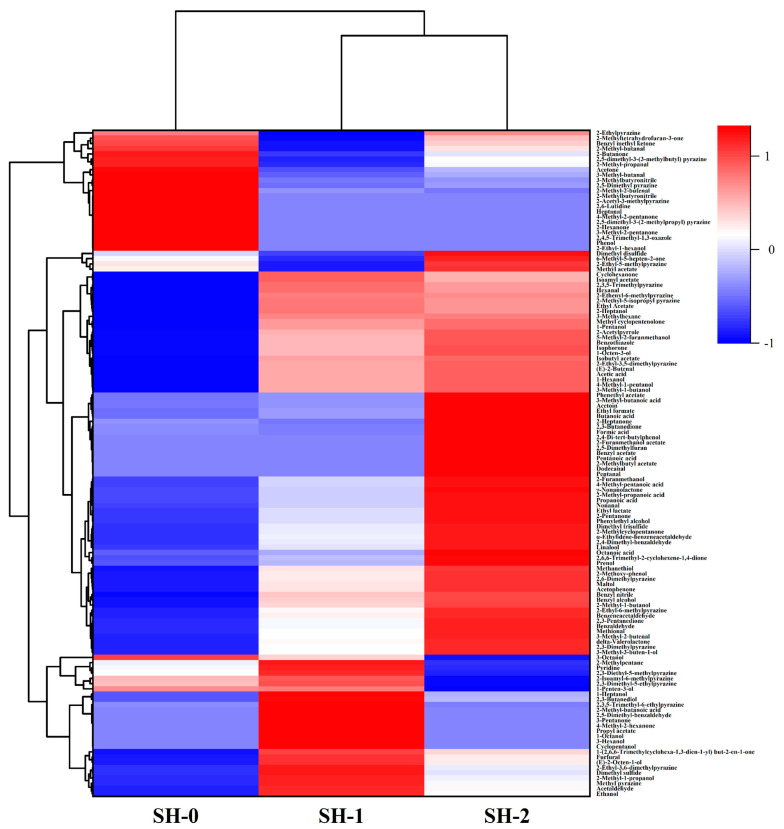
Heat map and hierarchical cluster analysis of the volatile flavor compounds in different soybean hydrolysate samples based on the GC-MS data after standardization. Darker red and darker blue respectively indicate higher and lower concentrations of volatile flavor compounds.

**Figure 3 foods-12-04513-f003:**
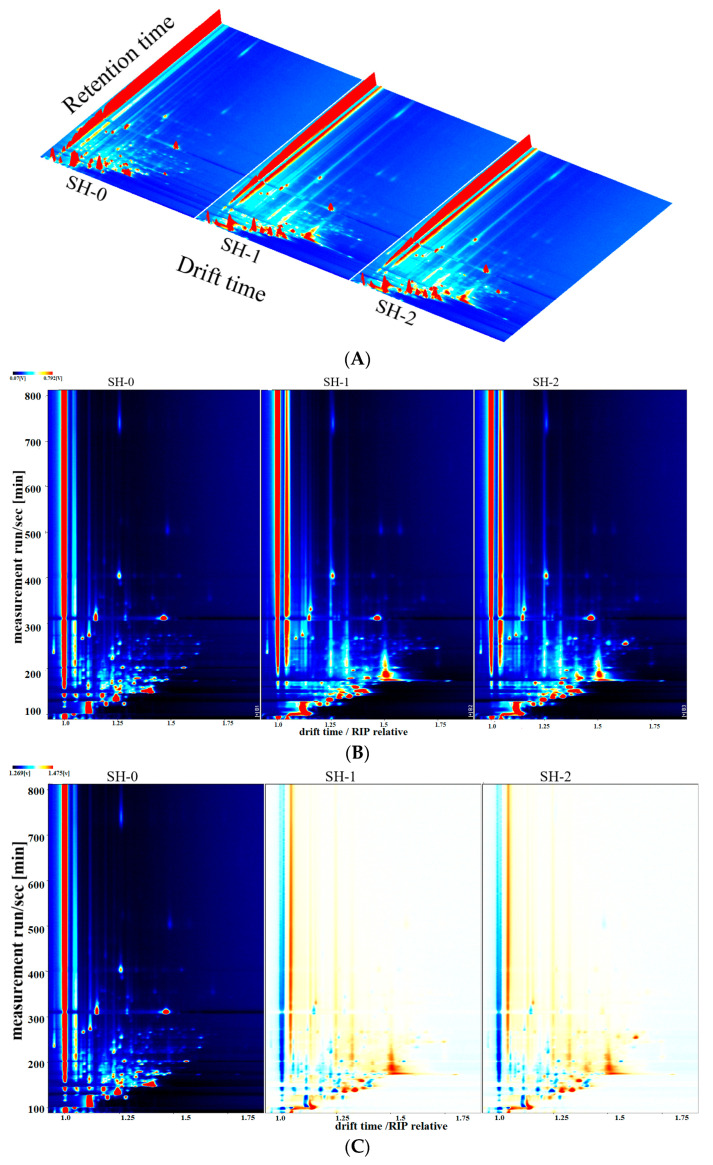
Topographic plots of soybean hydrolysate samples. Each colored dot on the spectrum represents a volatile compound, with red color indicating a high intensity and blue color indicating a low intensity. (**A**) 3D-topographic features of the volatile compounds (within the blue background, the red vertical lines on the left represent the reactive ion peaks); (**B**) 2D-topographic features of the volatile compounds; (**C**) 2D-topographic plot with a difference comparison model. Red and blue dots indicated that the concentrations of the compounds were higher or lower than the reference, respectively. The red vertical line at the abscissa 1.0 represents the RIP peak (Reaction Ion Peak, normalized); (**D**) Gallery plot fingerprints of the volatile compounds in soybean hydrolysate samples. Each dot represents a volatile flavor compound, with the intensity of its color indicating its concentration.

**Figure 4 foods-12-04513-f004:**
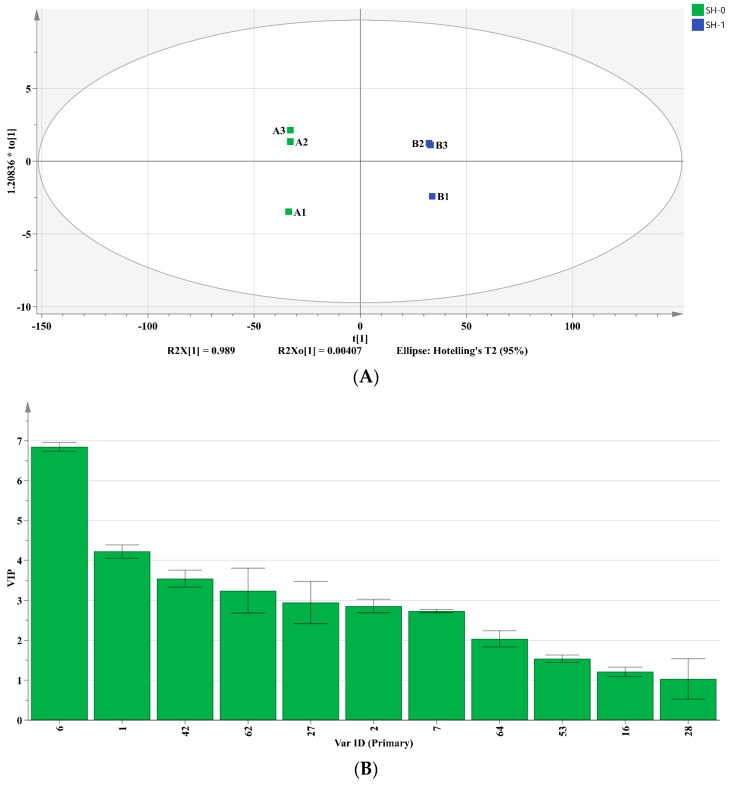
Orthogonal partial least squares analyses (OPLS-DAs) of the volatile compounds in soybean hydrolysate samples based on the GC-MS and GC-IMS data. The three OPLS-DA models are as follows: (**A**,**B**) SH-0 and SH-1; (**C**,**D**) SH-0 and SH-2; (**E**,**F**) SH-1 and SH-2. (**A**,**C**,**E**) represent the score plots, while (**B**,**D**,**F**) illustrate the variable importance in projection values for the respective models. A1, A2, and A3; B1, B2, and B3; and C1, C2, and C3 denote the three replicates of the SH-0, SH-1, and SH-2 groups, respectively. The Var ID numbers are consistent with the compound numbers in Appendix A.

**Figure 5 foods-12-04513-f005:**
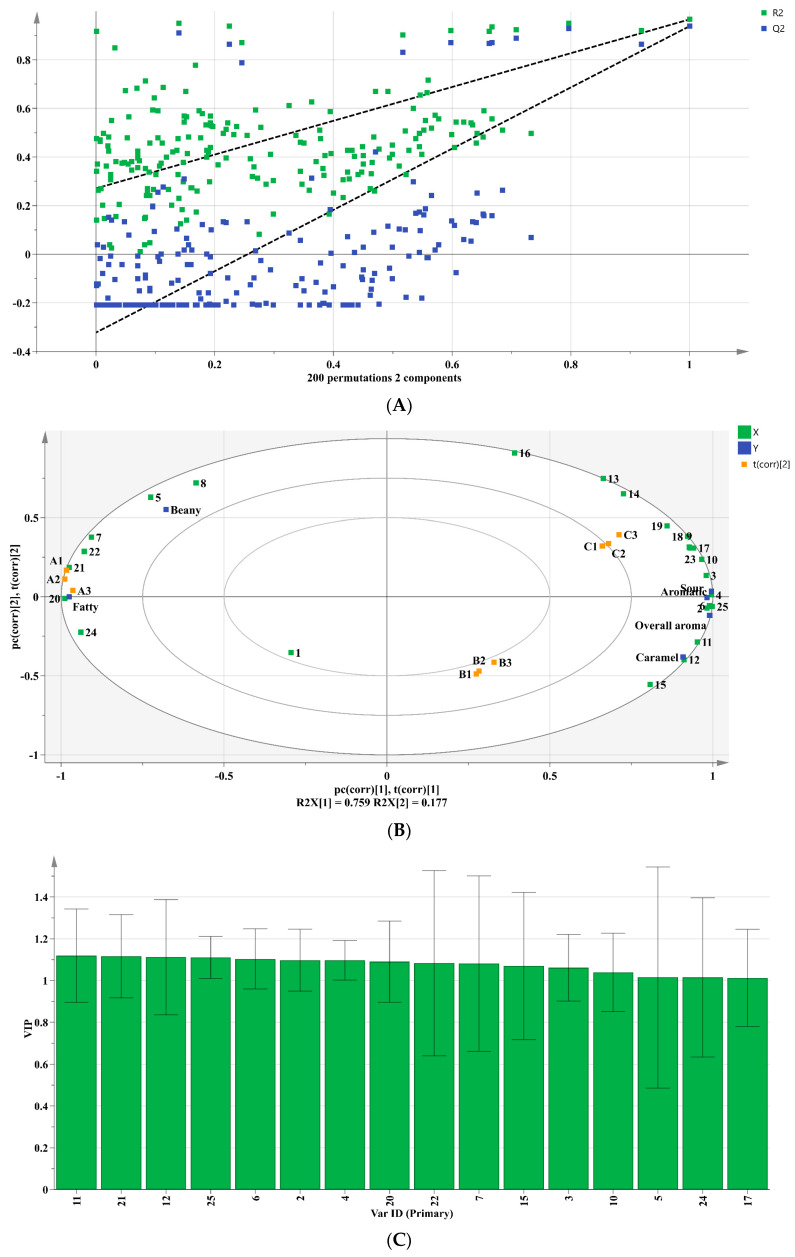
Partial least squares regression (PLSR) analyses of key volatile compounds. (**A**) Cross validation diagram (*n* = 200); (**B**) Biplot loadings; (**C**) variable importance in projection values; (**D**) observation and prediction diagram. A1, A2, and A3; B1, B2, and B3; and C1, C2, and C3 denote the three replicates of the SH-0, SH-1, and SH-2 groups, respectively. The Var ID numbers are consistent with the compound numbers in Table 1.

**Table 1 foods-12-04513-t001:** Relative odor activity values (ROAV) and odor characteristics of volatile compounds of soybean protein hydrolysates.

Number	Compounds	OT ^a^	SH-0	SH-1	SH-2	Identification	Odor Descriptions ^b^
1	1-Penten-3-ol	0.4	1.48	1.13	1.16	MS, RI	butter, fish, green
2	3-Methyl-1-butanol	220	0.01	2.74	3.60	MS, RI	whiskey, malt, burnt
3	2-Methyl-1-butanol	16	ND	5.99	10.18	MS, RI	malt, wine, onion
4	1-Octen-3-ol	1.5	0.35	12.96	18.72	MS, RI	mushroom, cheese, sweet
5	2-Methyl-propanal	1.5	11.83	6.24	8.26	MS, RI	fruit, caramel
6	(E)-2-Butenal	0.3	ND	1.00	1.34	MS, RI	pungent
7	3-Methyl-butanal	1.2	84.34	69.54	95.34	MS, RI	apple, fruit
8	2-Methyl-butanal	4.4	11.43	5.50	8.16	MS, RI	coffee, bread, banana
9	Benzaldehyde	3	7.47	10.01	16.01	MS, RI	almond, caramel
10	Benzeneacetaldehyde	4	0.56	1.03	1.66	MS, RI	fruit
11	Ethyl acetate	5	ND	6.10	6.19	MS, RI	pineapple, fruit
12	Isoamyl acetate	2	ND	1.88	1.65	MS, RI	banana
13	2,3-Butanedione	1	1.36	0.94	5.33	MS, RI	cream
14	Acetoin	14	0.44	0.54	2.71	MS, RI	cream
15	β-damascenone	0.002	100	100	100	MS, RI	apple. rose, honey
16	Dimethyl disulfide	1.1	1.16	0.31	2.15	MS, RI	onion, cabbage
17	Methional	1.4	1.19	2.26	4.00	MS, RI	cooked potato
18	Dimethyl trisulfide	0.1	3.26	9.40	20.80	MS, RI	sulfur, fish, cabbage
19	2-Furanmethanol	6	2.27	2.48	4.30	MS, RI	almond
20	Hexanal-D	4.5	9.17	5.33	3.83	IMS, RI	grass, tallow, fat
21	Hexanal-M	4.5	9.32	1.56	1.25	IMS, RI	grass, tallow, fat
22	3-Methylthiopropanal-M	1.4	27.67	22.09	21.76	IMS, RI	meat, onion, fruit
23	3-Methylthiopropanal-D	1.4	7.44	11.31	14.37	IMS, RI	meat, onion, fruit
24	Nonanal	1	21.99	15.12	8.85	IMS, RI	fat, citrus, green
25	2-Pentanone	10	3.59	15.69	17.63	IMS, RI	fruit, pungent

Not detected in sample. ^a^ OT: odor threshold value as reported in the literature reference. ^b^ Odor descriptions were cited from www.flavornet.org and www.femaflavor.org (accessed on 12 December 2023).

## Data Availability

Data is contained within the article or Appendix A.

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
