# Peer review of "Effects of Fermentation with Tetragenococcus halophilus and Zygosaccharomyces rouxii on the Volatile Profiles of Soybean Protein Hydrolysates"

_foods, 2023, doi:10.3390/foods12244513_

Round 1

Reviewer 1 Report

Comments and Suggestions for Authors

Dear Editors and authors, 

Major comments

1-What is the reason for mixing bacteria with yeast? It is not better to add each one separately before mixing them. What is the greater influence, bacteria or yeast? Who produced these compounds?

Minor comments 

1. I suggested changing the title of the manuscript to: Effects of fermentation with Tetragenococcus halophilus and Zygosaccharomyces rouxii on the volatile profiles of soybean protein hydrolysates.

2.The summary of the manuscript needs to add some modifications to be more clear to the reader. Such as adding some results numbers and the best final conclusion. 

3.It is suggested to support the presenter with some references that are similar to this study, such as:

Niamah, A. K., Al-fekaiki, D. F., Al-Sahlany, S. T. G., Verma, D. K., Patel, A. R., & Singh, S. (2023). Investigating the effect of addition of probiotic microorganisms (bacteria or yeast) to yoghurt on the viability and volatile aromatic profiles. Journal of Food Measurement & Characterization, 17(5), 5463-5473.

Liu, N., Li, X., Hu, Y., Qin, L., Bao, A., Qin, W., & Miao, S. (2023). Effects of Lentilactobacillus buchneri and Kazachstania bulderi on the Quality and Flavor of Guizhou Fermented Red Sour Soup. Foods, 12(20), 3753.

4. Some of the work methods do not contain scientific references. Scientific references must be added to them that explain to us the methods used in the study, such as: Microorganisms and growth conditions and Preparation of enzymatic hydrolysates of defatted soybean flour

5. Line 108, Preparation of enzymatic hydrolysates of defatted soybean flour, This method is ambiguous and unclear. The authors did not mention the volume of the analyzed soybean sample, and how much starter cultures was added to it ? Was the addition of bacteria and yeast in a ratio of 1:1?

6. Line 165, Sensory evaluation, Sensory assessment conducted by 10 people? Where did they receive their training, what qualifications do they have?

7. Why were the numbers of bacteria and yeast not calculated in the final product?

8. The conclusions must be rewritten. The conclusions contain many results. Here, there is no chapter of the results. You should write the conclusions, not the results.

Reviewer 2 Report

Comments and Suggestions for Authors

Comments on Manuscript Foods-2746913 "Effects of fermentation with lactic acid bacteria and yeast on the volatile profiles of soybean protein hydrolysates"

In this paper, the authors studied the effects of fermentation with lactic acid bacteria (LAB) and yeast on the volatile flavor compounds (VFCs) of different soybean protein hydrolysates. Modern techniques – GC-MS and GC-IMS – were used for the analysis of VFCs. The topic of investigations is actual for food science and technology. The findings provide a theoretical basis for improving and assessing the flavor of soybean protein hydrolysates

The Abstract reflect the essence of the paper. Methods and materials correspond to the aim of the research presented. The discussion and conclusions arise from the obtained data and their interpretation. However, the methodological part and the presentation of the results require improvement and need to be revised.  My comments and suggestions are in the attachment.

Round 2

Reviewer 1 Report

Comments and Suggestions for Authors

Dear Editors, 

The authors made all necessary changes to improve the manuscript, and now I recommend it for publication in its current form.

Author Response

Dear Reviewer:

Thank you for your constructive feedback and for recommending our manuscript for publication. We are grateful for the opportunity to improve our work based on your valuable insights. Your positive evaluation of the revised manuscript is highly encouraging. 

We appreciate the time and effort you dedicated to reviewing our work. We believe that the changes made have indeed strengthened the manuscript, and we are pleased to hear that you find it suitable for publication in its current form. 

Thank you again for your support and guidance throughout the review process. 

Best regards

Guowan Su

Reviewer 2 Report

Comments and Suggestions for Authors

Please see the attachment for details.
